# Adipocytes in the Uterine Wall during Experimental Healing and in Cesarean Scars during Pregnancy

**DOI:** 10.3390/ijms242015255

**Published:** 2023-10-17

**Authors:** Natalia Tikhonova, Andrey P. Milovanov, Valentina V. Aleksankina, Ilyas A. Kulikov, Tatiana V. Fokina, Andrey P. Aleksankin, Tamara N. Belousova, Ludmila M. Mikhaleva, Natalya V. Niziaeva

**Affiliations:** 1Avtsyn Research Institute of Human Morphology of FSBSI “Petrovsky National Research Centre of Surgery”, 119991 Moscow, Russiatatyana-doc-6@mail.ru (T.V.F.); aap2004@yandex.ru (A.P.A.); niziaeva@gmail.com (N.V.N.); 2SBHI of the Moscow Region “Vidnovsky Perinatal Center”, 142700 Moscow, Russiabeltamara1@mail.ru (T.N.B.)

**Keywords:** caesarean section, uterine scar, adipocytes, macrophages, uterine wall healing, placenta accreta spectrum, Sprague–Dawley, uterus

## Abstract

We have suggested that adipocytes in uterine scars may affect the development of the placenta accrete spectrum (PAS). In the experimental part, we explored adipocytes in the uterine wall by the twelfth sexual cycle after surgery. In the clinical part, we investigated adipocyte clusters in the cesarean scar of pregnant women with and without PAS. The uterine wall was evaluated in gross and histological sections using morphometry, histochemistry (hematoxylin and eosin stain, Mallory stain), and immunohistochemistry for FABP4 (adipocyte markers), CD68, CD163, CD206 (macrophages), CD 34 (endothelium), cytokeratin 8 (epithelium), aSMA (smooth muscle cells). The design included an experimental study on Sprague–Dawley rats (n = 18) after a full-thickness surgical incision on the seventh (n = 6), 30th (n = 6), and 60th day (n = 6). The clinical groups include pregnant women without uterine scars (n = 10), pregnant women with a uterine scar after previous cesarean sections (n = 10), and women with PAS (n = 11). Statistical processing was carried out using nonparametric methods. Comparisons were conducted using the Mann–Whitney U-test and Kruskal–Wallis test. Statistical significance was considered at *p* < 0.05. On the seventh day, the rat uterine horn was enveloped by adipose tissue, which contained crown-like structures with FABP4+, CD68+, CD206+, and CD163+ cells. FABP4+ cells in the uterine wall were absent by the 30th day. The number of CD206+ and CD163+ cells in the adipose tissue decreased by the 30th day. On the 60th day, the attachment of fat tissue was revealed in the form of single strands. The serous layer around the damaged area totally recovered on the 60th day. FABP4+ cells were not detected in the uterine wall samples from pregnant women without a previous cesarean section. Adipocytes were found in the scar during non-complicated pregnancy and with PAS. Reducing the number of CD68+ cells in adipocyte clusters, there were in myometrium with PAS. Increased CD206+ and CD163+ cells were revealed in uterine adipocyte clusters of the group. According to the experimental finding, adipocytes should be absent in the uterine wall by the 12th sexual cycle after a full-thickness surgical incision. The presence of adipocyte clusters in cesarean scar indicated the disturbance of cell interaction. Differences in the numbers of CD206 and CD163 cells in adipocyte clusters between groups with and without PAS may be indirect evidence that uterine adipocytes affect the development of PAS.

## 1. Introduction

Currently, adipose tissue has been regarded as a distinct endocrine organ that regulates many aspects of body physiology, including energy maintenance, insulin sensitivity, body temperature, and immune responses. The most important property of adipose tissue is its high degree of plasticity. Physiological stimuli cause dramatic changes in the metabolism, structure, and phenotype of adipose tissue to meet the needs of the body [1,2].

In 2020, Shook BA et al. [3] revealed that direct dermal adipocytes are involved in skin wound healing. The role of endogenous adipose tissue in the restoration of the uterine wall after damage is not well researched, although some authors published data about the positive effect of cell fraction from adipose tissue on the repair of the damaged uterine wall in laboratory rodents [4,5]. Mesenchymal cells derived from adipose tissue are the most promising cell types in regenerative medicine and gynecology [5,6]. The interaction of the components of the visceral fat depot with a damaged uterine wall in rats is also known [7,8].

The adipocytes in the uterine wall of a woman may occur in various pathologies such as diffuse chronic endometritis [9], uterine lipoleiomyoma [10], and angiomyolipomas [11]. On the other hand, mesenchymal stromal cells in the uterine wall of mice and humans can differentiate into adipocytes and other cell lines, including the formation of new vessels. Moreover, adult myometrial pluripotential precursors promote the restoration of the architecture of the damaged myometrium [12]. We have not found any research devoted to the adipocyte component for uterine wall restoration and scar formation after a cesarean section in women. However, recently, we revealed adipocyte component in cesarean scar during uncomplicated pregnancy and pregnancy with abnormal invasion of the placenta [13].

Abnormal invasion of the placenta or placenta accrete spectrum (PAS) is a complex obstetric complication associated with massive bleeding and may require urgent hysterectomy [14]. Three subtypes of PAS are distinguished: (1) superficial placenta accreta, where the villi attach directly to the surface of the myometrium without invading it; (2) placenta increta, in which case the villi penetrate deep into the myometrium up to the outer layer; and (3) placenta percreta: an invasive trophoblast penetrates the serosa of the uterus. The term PAS includes all these subtypes [15].

Despite numerous studies, the pathogenesis of PAS remains unknown. Abnormal invasion of the placental villi is mainly the result of changes in the structure of the uterine wall after surgery, especially after a cesarean section. The myometrium lesions are associated with collagen and fibrin depositions in and around the wound area, disruption of the structure of myofibrils, tissue edema, inflammation, and elastosis. Large defects after cesarean section contribute to pathological decidualization and allow placental anchor villi to penetrate through the decidua basalis into the underlying uterine myometrium [16,17].

Most patients with PAS have a history of abnormalities in the uterine cavity, such as a previous cesarean section or other surgical interventions on the uterus (myomectomy, operative hysteroscopy, etc.) [18,19]. Thus, the cesarean section and curettage are the main predisposing factors for PAS [20,21]. The risk of PAS increases with the number of previous cesarean sections [22] and the size of myometrial defect [23,24].

The uterine scar is a necessary but not enough factor for PAS. Identifying the distinctive features of uterine scars associated with the development of PAS will facilitate early diagnosis and treatment. The lack of animal models restricts studying the mechanisms for PAS genesis [25]. Therefore, clinical samples obtained during operative delivery are the primary source of information about the structure of the uteroplacental zone in PAS. In contrast to the absence of animal models for PAS, there is an available opportunity to study the mechanisms of uterine wall healing in a laboratory rodent model. In previous studies, we discovered that adipose tissue was involved in the early healing of the uterine wall of rats [7,8]. In addition, we found adipocytes in the uterine scars of pregnant women [13]. No other scientific publications are available to us that describe adipocytes in uterine scars, both after cesarean section and with abnormal invasion of chorionic villi. However, dermal adipocytes and macrophages are actively involved in healing skin wounds [26,27,28].

We have suggested that adipocytes in uterine scars may affect the development of PAS. Thus, on the one hand, we decided to investigate the interaction of fat tissue with the injured uterus in the rat model during a prolonged healing period. Our objective was to determine if adipocytes were present in the uterine wall by the twelfth sexual cycle following uterine surgery. On the other hand, we attempted to investigate the cell environment of adipocytes in the cesarean scar in pregnant women with and without PAS.

In this regard, we have assumed this to be a pilot study and chose immunohistochemical markers of FABP4 for adipocytes, CD206 and CD163 for the anti-inflammatory population of M2 macrophages involved in the regulation of adipose tissue homeostasis [29], as well as the traditional endothelial marker CD34 for studies of perivascular localization.

## 2. Results

### 2.1. Experimental Study

On the seventh day after surgery, the damaged area was completely covered with adipose tissue (Figure 1(A1,B4)). The contractility of the operated uterine horn was impaired by 30%, and the ratio of the operated horn length to the non-operated one was 1.3 (Figure 1(C1,C4)). On the 30th day, the coverage of adipose tissue was significantly reduced to 33% (Figure 1(B2,B4)). On the 60th day, it was reduced to 3% (Figure 1(B3,B4)) and detected as a single strand. The serous membrane around the damaged area has fully recovered by the 60th day (Figure 1). The contractility of the operated horn improved by the 30th day (Figure 1(C2,C4)), and it completely recovered by the 60th day (Figure 1(C3,C4)).

Histological examination on the seventh day revealed groups of FABP4+ and CD68+ cells in the healing zone and under the serous membrane. The damaged area of the horn was enveloped in a clutch of adipose tissue. The absence of a serous membrane and the attachment of adipocytes to the healing zone were revealed (Figure 2(A1)). A significant number of crown-like structures were present in the area of attachment of fat tissue (Figure 2(A1,B1,C1,D1,E1,F1,G1)). The crown-like structures disappeared by the 30th day (Figure 2(A2,B2,C2,D2,E2,F2,G2)). By the 30th day, the healing area in half of the animals remained covered with adipose tissue. However, a connective tissue strip was revealed between the uterine wall and adipocytes in all animals (Figure 2(A2,B2,C2,D2,E2,F2,G2)). By the 60th day, adipose tissue attachment to the incision area was not detected, and residual attachment of adipose tissue to the uterine horn was observed outside the incision area (Figure 2(A3,A4)).

Immunohistochemical examination on the attachment of adipose tissue to the healing zone revealed a decrease in the content of CD68+, CD206+, and CD163+ cells by the 30th day. The difference between the 30th and 60th days, according to these markers in the adipose tissue attached to the injury zone and outside the injury zone, was not revealed (Figure 2(H2–H5)).

### 2.2. Study Samples of Uterine Wall after Caesarean Section from Women

The retrospective study included samples from the uterine wall region from 10 women with no history of cesarean section (CS), no signs of the uterine scar (control; gestation 39–40 weeks, age 25–41 years), 10 women with the cesarean scar (scar; 1–4 CS, gestation 38–40 weeks, age 27–41 years), and 11 women with PAS (PAS; 2–5 CS, gestation 27–38 weeks, age 29–42 years) (Table 1). The criteria for inclusion in the group with the first CS (no history of previous CS) were operative delivery, such as the incorrect position of the fetus, anatomically narrow pelvis, fetal macrosomia, the presence of a uterine scar, and eyes pathology in women (such as pathology of the retina and myopia). The exclusion criterion for this group was the history of instrumental damage to the uterine wall.

Exclusion criteria for all groups included infectious diseases, autoimmune pathologies, metabolic disturbances, obesity of the second and third degree, and diabetes mellitus type 2.

All women with PAS included in the study had a history of at least one cesarean section. The age of the patients ranged from 29 to 42 years, and the operative delivery period was from 27 to 38 weeks of gestation; the number of previous abdominal deliveries by cesarean section was from 1 to 3 (Table 1). Placenta was diagnosed in five cases as accreta (45%) and in 6 cases—placenta increta (55%).

The uterine wall samples from women without previous CS did not contain scar tissue. They were represented by compact fragments of the myometrium with densely arranged bundles of myocytes (Figure 3(A1)). Extravillous cytotrophoblast invasion was detected in the radial arteries (Figure 3(B1)). There was no positive reaction for FABP4 during immunohistochemical examination (Figure 3(D1)).

Adipocyte fields were found in the samples from women with a cesarean scar in both groups (Figure 3(A2,A3,B2,B3,D2,D3)). Scar tissue separating individual muscle bundles and forming larger fields was detected in the uterine wall (Figure 3(A2,A3,B2,B3)). Adipocyte clusters of different sizes were found only in the scar area (Figure 3(A2,A3,B2,B3,D2,D3)). Localization of the adipocyte clusters in scar tissue was noted in the perimetrium and in the perivascular zone (Figure 3(A2,A3,B2,B3,D2,D3)). The size of the adipocyte clusters varied greatly from groups including two cells to large areas stretching along the vessels and along the uterine wall in the serous layer. The number of adipocyte clusters in the group with PAS containing at least three cells when stained with FABP4 was greater than in the group with a scar and non-complicated pregnancy (*p* < 0.05) (Figure 3(E2)). No adipocytes were found in the uterine wall outside the scar tissue and in the control group (Figure 3A). In the uterine wall outside the adipocyte clusters, a decrease in CD68 cells was detected in the groups with a scar compared to the control group (*p* < 0.05) (Figure 3(E2)).

The study of CD68, CD163, and CD206 in the zones of adipocytic clusters revealed a decrease in the number of CD68+ cells but an increase in CD163+ and CD206+ in the PAS group (Figure 4).

Histological analysis of uteroplacental region in women with PAS revealed extensive scar components penetrating all uterine layers, thinning of the uterine wall, fragmentation and separation of myocyte bundles by sclerosis fields, and tissue edema (Figure 5A–C).

In six samples (55%), deeply implanted villi were found and classified as placenta increta (Figure 5A–E). In five samples (45%), the villous tissue reached the muscle layer but did not penetrate it. These samples are classified as placenta accreta (Figure 5F). No villi penetrated the entire thickness of the uterine wall (placenta percreta). There were vast intervillous thrombi and single infarcts of villous tree. Vast fibrinoid deposits were found around all deeply implanted villi, separating them from the surrounding myometrium. In the fibrinoid deposits area, the uteroplacental border appears to be wave (Figure 5A–C).

In the uterine scar area, there were disturbances of the smooth muscle structure. The vessels of the scar zone were characterized by a narrowed lumen and thickened wall (Figure 5B and Figure 6A,B).

Under the zone of attachment and invasion of placental villi, adipocyte clusters in all cases were found. Groups of adipocytes were located predominantly in the perivascular zone of the scar tissue in groups of 3 to 50 cells (Figure 6B and Figure 7A–C) or in the perimetrium in groups of 50 cells and more, which, as a rule, stretched along the serous lining of the uterine wall (Figure 5C). The adipocyte group thickness located in the serosa was 3–6 cells, and the length corresponded to the PAS zone, but in all cases, it was limited to the scar tissue (Figure 5A,C). However, in the area of the scar between the disconnected smooth muscle fibers, there were also single adipocytes (Figure 7A, black arrow). The average size of cells in a group depended on the size of the group; the larger cells were found in the larger clusters. A single FABP 4 + cell size with a vast vacuole occupying more than half of the cell area ranged from 566 µm^2^ to 775 µm^2^ on average 587 (576;681) µm^2^, and the size of cells in groups of up to 2 to 5 was 1640 (1450;1685) µm^2^. The average cell size in groups of more than five adipocytes near the vessels was 1460 (1335;1655) µm^2^; the minimum size was 167 µm^2^, and the maximum one was 1950 µm^2^. The average size of adipocytes in the perimetrium was 1260 (579;5020) mm^2^ with a minimum size of 103 mm^2^ and a maximum of 5960 mm^2^. The size of single cells was significantly less than the average size of cells organized into clusters (*p* < 0.05). The mean cell size in clusters did not differ in cluster size significantly (*p* > 0.05). The size of single cells was reliably smaller than the average size of adipose cell cluster (*p* < 0.05). Tendencies to the formation of lobules, in which adipocytes were organized in adipose tissue, were not observed in our study.

Long adipocyte streaks along the serosa were located under the muscular layer and often accompanied by blood vessels. The vessels are arranged in parallel to the serosa (Figure 8A,B). In some fields of micropreparations, several vessels (both venous and arterial) were noted in the fat layers of the perimetrium placed perpendicular to the section but not longitudinally (Figure 8C). Examining invasive extravillous trophoblast cells in the PAS zone did not reveal the penetration of trophoblast cells into the adipocyte areas or invasion through the adipocyte fields via the uterine serosa (Figure 8C,D).

The vascular component examination in the clusters of fat cells revealed many small vessels lined with CD 34 endothelial cells (Figure 9A,B). Around the muscular wall of the arteries, the most closely located were quite small diameter blood vessels. Some of them had collapsed lumen, and they did not contain erythrocytes (Figure 9A). In these adipocyte clusters, there was a vascular component between the adipose cells. Small vessels are arranged more abundantly around arteries and the largest vessels, especially close to ones with red blood cells in the lumen (Figure 9B).

Immunohistochemical staining with primary antibodies to CD163 allowed us to detect that CD163+ cell clusters are located around the vascular walls, and their number depends on the diameter of the vessel. CD163+ cells are not detected in adipocyte clusters outside the vessels (Figure 9E,F). CD206 cells were found both inside and outside the vascular lumen, including the endothelial layer. They infiltrate more adipocyte fields compared to the perivascular zone (*p* < 0.05) and surrounding tissue (*p* < 0.05) (Figure 9G,H).

In the scar area in cases of PAS, the villus penetrated into the large vessel (Figure 10A), retained the endothelial lining (Figure 10B,C), and contained erythrocytes.

Of note, in the tissue between the lumen of the vascular collectors, there was a paucity group containing small FABP 4+ CD 34- cells with vast single cytoplasmic vacuoles. In our view, these cells may be attributed to the adipocyte lineage.

## 3. Discussion

Our previous study using a rat model investigated morphological changes in the uterine horn wall during the healing over a 2-week period after a full-thickness surgical incision. On the fourth day, there was a tight attachment of the intraabdominal adipose tissue to the site of damage simultaneously with the complete disappearance of the wound channel [7]. Later, we demonstrated that cells of adipocyte origin are actively involved in the process of healing through damage to the uterine wall of rats at the earliest stages. The adipose tissue attached to the wound defect created an isolating tissue layer separating the damaged area from the abdominal cavity. It was presumably the main or additional source for entering pluripotent and immune cells in this area to restore impaired tissues. The period of the most active interaction between adipose tissue and the damaged horn started from the third day and continued until the end of observation—15 days. In this period, the activation of macrophages in the healing zone was also observed. However, in areas with active inflammation, neutrophils and adipocytes were absent. The study of the further participation of the adipocyte component in the regenerating process later than 15 days after the operation seems to be relevant in connection to an unresolved problem of uterine defective scar forming after cesarean section in obstetric practice and being the cause of many severe complications during pregnancy [8].

In this study, we showed that the adipocyte component of the rat uterus disappears at the late stages of the healing after the surgical incision (60 days, the 12th estrous cycle). The adipocyte component was detected in the early period [8]. The crown-like structures disappear by the 30th day. Some researchers reported that crown-like structures are associated with adipocyte lipolysis, the release of lipids from adipocytes, and an inflammatory reaction in white adipose tissue [30,31,32]. CD206 is a mannose receptor that is present in M2 macrophages, immature dendritic cells, and some endothelial cells. It is involved in the internalization and degradation of collagen [33]. One of the key roles of CD206 is the regulation of the level of inflammatory products [34]. CD206 is expressed at low levels during inflammation and at high levels during inflammation resolution [35]. Thus, it can be assumed that CD206 cells dispose of the products of the attenuating inflammation in the zone of adipose tissue attachment to the damaged uterine wall, and crown-like structures indicate an active process of lipolysis on the seventh day after surgery.

In addition, the accumulation of CD206 and/or CD163 macrophages correlates with fibrosis [36,37]. In our study, by the 30th day after the operation, the crown-like structures disappeared, and the number of CD206 and CD163 cells decreased. Moreover, a connective tissue band appeared, separating the uterine wall from the attached adipose tissue. Possibly, the formation of this connective tissue strip begins on the seventh day, and lipolysis is necessary for the energy supply of these processes.

The finding of adipocyte clusters in the uterine wall of pregnant women is quite an unusual phenomenon. In a normal uterus, such facts are not noted. According to the literature data, adipocytes, together with macrophages, are involved in cellular debris clearance as well as cytokines and growth factors secretion in skin wound healing [26]. The absence of adipocytes in the healing zone results in a decrease in fibroblast recruitment and subsequent disturbances in the extracellular matrix remodeling [27]. In addition, adipocytes have been found in human skin granulation tissue. The frequency of detection and the structure of adipocytes change depending on the type of wound, anatomical origin, and age of the wound [28].

Considering the role of adipocytes in wound healing, we assume that there are two ways to appear adipocyte clusters in the uteroplacental region with PAS. In the first case, the identified adipocyte clusters formed at the stage of wound healing after the cesarean section and, possibly, are associated with an impairment of the tissue repair process during that period. Accompanying adipocyte clusters close to uterine vessels does not contradict this hypothesis. In the second case, adipocyte presence may be associated with the reaction of the uterine wall to the abnormal invasion of placental villi. This may be evidenced by our finding of single or small groups of FABP4+ cells. Adipocyte clusters were located around vessels of medium diameter.

In the perimetrium, the adipocyte clusters, stretching along the edge of the uterine wall, were much larger and longer than in the myometrium. Often, they accompany vessels located along the uterine wall (Figure 5C,D and Figure 8A,B). Single adipocytes in groups of 2–4 cells were in the scar tissue (Figure 3A) or between the vascular collectors (Figure 10D). Ooi G et al. revealed CD34, traditionally considered the marker for endothelium, on the precursors of white adipocytes [38]. Moreover, endothelial cells of capillaries and arteries can synthesize FABP4, and arterial endothelial cells acquire this ability in response to endovascular damage to the vessel [39]. Therefore, the escort of arteries by adipocytes that we found could also be a consequence of damage during trophoblastic invasion (Figure 8 and Figure 9).

The question of cell interaction between the extravillous invasive cytotrophoblast and adipocytes remains unclear. We have not seen the penetration of cytotrophoblast cells into the layer of adipocytes. In one case, the cytotrophoblast approached the band of adipocytes located along the serosa but did not penetrate it (Figure 8C,D). Perhaps the layer of adipocytes prevented deeper trophoblastic invasion. We did not find significant differences in the number of adipocyte clusters between the groups with a scar, but we noted a tendency to increase adipocyte clusters in samples with PAS (Figure 3(E2)). Perhaps such results are associated with a small sample size.

Macrophages are the key regulators for the wound healing process [40], stimulate adipogenesis in granulation tissue [28], and play a significant role in fat depot homeostasis [41,42]. The inflammatory macrophages suppress the differentiation of adipocytes, preventing the synthesis of endogenous fatty acids. The appearance of adipocytes in the wound tissue indicates a predominant anti-inflammatory environment and may be associated with improved healing outcomes [28]. The decrease in the number of CD68 cells in the myometrium of the groups with scars compared to the control group may be related to the impact of cell crosstalk in the scar. An increase in CD206 and CD163 in adipocyte clusters for the group with PAS compared to the group with a scar and non-complicated pregnancy may indicate a rise in inflammatory products. 

The perivascular localization of CD163+ cells revealed in this work suggests that they can regulate vascular function. CD163 has been identified as a haptoglobin-hemoglobin receptor and exclusively in cells of monocyte-macrophage origin [43]. In vitro, CD 163 expression may be suppressed by pro-inflammatory mediators, whereas the anti-inflammatory cytokine interleukin-10 activates CD163 [44]. In vivo, CD 163-positive macrophages and soluble CD 163 are found in the late acute and chronic phases of inflammation [45]. CD 163 macrophages are classified as perivascular macrophages capable of absorbing hemoglobin, stimulating angiogenesis, and regulating vascular permeability [46]. In this study, CD 163+ cells in fatty layers also clustered around arterial and venous vessels.

We noted a more intensive infiltration of adipocyte clusters by CD206 cells compared to the surrounding tissue. CD206 is expressed mainly by tissue macrophages, dendritic and endothelial cells [47].

Macrophages CD 206+ and CD 163+ belong to the M2 anti-inflammatory population. However, an increase in CD206 and CD163 cells in adipocyte clusters of PAS patients may indicate an active inflammatory reaction and the presence of inflammatory products (Figure 4(B3,C3)). That is supported by the experimental study, where we revealed an increase in CD206 and CD163 cells on the seventh day compared to the 30th and 60th days of rat uterine healing.

## 4. Materials and Methods

Experimental study. Animals. The incision through the right uterine horn was performed on female Sprague–Dawley rats (n = 18; age 14–16 weeks, weight 220–280 g). All animal studies were performed in accordance with Directive 2010/63/EU of the European Parliament, the Council of the European Union on the protection of animals used for scientific purposes, GOST 33216-2014 “Rules for working with rodents and rabbits” and were approved by Bioethical Committee of the Research Institute of Human Morphology (protocol No 35 (11) dated 23 March 2022). Rats were housed in clean barrier facilities under humidity and light-controlled conditions and fed ad libitum with free access to water. The operation was performed in the estrous phase of the reproductive cycle. To determine the phase of the estrous cycle, vaginal smears stained using Diahim-Diff-Quick (NPF ABRIS+, Ltd., cat. No. 451) were examined [48]. The levels of estradiol and progesterone in the blood serum were determined by enzyme immunoassay using the ImmunoFA-Estradiol and ImmunoFA-PG reagent kits (CJSC NVO Immunotech, cat. No. IP02-03 and IP02-08), respectively [49].

Surgical intervention. An incision was made via the wall of the right uterine horn in the estrus phase. The left uterine horn was intact for comparative control. The operation was performed with general anesthesia support, the anesthetic “Zoletil “ (Vibrac, France)—10 mg/kg and the sedative drug “ Xylanit “ (LLC “Nita-Pharm”, Moscow, Russia)—0.1 mg/kg were administered by intramuscular injection. The skin and underlying tissue layers of the anterior abdominal wall were dissected along the linea alba. The right uterine horn was mobilized, and a longitudinal incision with a length of 10 mm was made on the antimesometrial side. The uterus and abdominal wall were sutured by a continuous wrapping suture with Ultrasorb USP 6/0 and the skin by a U-shaped knot suture with Monosorb USP 4/0. The sutured surgical site was treated with 0.05% chlorhexidine bigluconate (YuzhFarm, LLC., Krasnodar, Russia) and Terramycin Aerosol (IGS Aerosol GmbH, Wehr, Germany).

Morphological study. The animals were sacrificed under ether anesthesia. Gross morphology observation, histological, and immunohistochemical assay were performed on the 7th, 30th, and 60th days after the operation. The damaged and adjacent areas of the operated horn and the view of the unoperated horn were analyzed. The degree of adhesions between the uterine wall and adipose tissue was calculated as a percentage. In cases of envelopment of the adipose tissue completely of the operated uterine horn segment, the degree of the adhesions was taken as 100%. After the removal of the fat tissue coupling and the attached fat tissue cords, the contractility of the right operated and left intact horns were compared. Horns were evaluated 10 min after the removal of samples. The change in the right-operated horn contractility was expressed as the ratio of the right horn length to the left horn length. All manipulations were carried out after the complete contraction of both horns. For histological examination, tissue samples were fixed with 10% buffered formalin (#60-001/S, LLC «BioVitrum», St. Petersburg, Russia). After 48 h of fixation, the material was washed, dehydrated, and embedded in Histomix Extra paraffin (#10342, LLC «BioVitrum», Russia). Sections from paraffin blocks with a thickness of 4 μm were obtained on a Sakura rotary microtome Accu-Cut SRM200 (Sakura Finetek Europe B.V., Flemingweg, The Netherlands), stained with hematoxylin and eosin (#07-006, LLC «BioVitrum», Russia), Mallory trichrome kit (#21-036, LLC «BioVitrum», Russia).

Immunohistochemical study. Adipocytes were detected with rabbit monoclonal antibodies (mAb) to FABP4 (#ab219595, Abcam, San Francisco, CA, USA), a transporter protein of lipids in adipocytes; macrophages CD68+ with mouse mAb (#ab31630, Abcam), CD206+ with rabbit polyclonal antibodies (#ab64693, Abcam), CD163+ with rabbit mAb (#ab182422, Abcam), endothelial cells CD34+ with rabbit mAb (#ab81289, Abcаm), smooth muscle cells α-SMA+ with mouse mAb (#ab7817, Abcam).

Clinical study. The retrospective study included samples from the uteroplacental region taken from women with no uterine scar and previous cesarean sections (n = 10). The study included groups of women with uterine scars with previous cesarean sections (n = 10) and a group with PAS (n = 10). Samples were received after abdominal delivery at the State Budgetary Health Institution of the Moscow Region” Vidnovsky Perinatal Center,” Vidnoye, Moscow Region, Russia, from January to December 2021. All patients were managed according to local protocols, including conservative surgical management. All procedures performed in this study involving all patients comply with the ethical standards of the institution’s ethics committee and the 1964 Declaration of Helsinki and its subsequent amendments or comparable ethical standards. This research was approved by the Bioethics Committee at FSBI (protocol No 35 (11) dated 23 March 2022). All included women gave voluntary agreements to participate in the study in accordance with the Declaration of Helsinki on the conduct of biomedical research. Previously, histological sections were examined to reveal an adipocyte tissue component, and 11 samples were selected. Paraffin blocks for the manufacture of micropreparations were obtained from biobanking. 

Histological examination. Sections from paraffin-embedded blocks 4 µm thick were obtained using the rotary microtome Accu-Cut SRM200 (Sakura, Japan). Microsections placed on glass slides were deparaffinized and rehydrated in a graded ethanol series, then washed in water and stained with hematoxylin and eosin (LLC «BioVitrum», article: 07-006) or Mallory trichrome kit (#21-036, LLC «BioVitrum», Russia) for detection of connective tissue according to Mallory (LLC «BioVitrum», article: 21-036), then again they were dehydrated and placed in Vitrogel (LLC «BioVitrum», article 12-005) for further microscopic examination.

Immunohistochemical study. Sections from the paraffin blocks were mounted on lysine-coated glass slides (Menzel-Glaser Polysine^®^), rehydrated, followed by heat-mediated antigen retrieval in citrate solution (pH 6.0), blocked [1 h at RT in 10% goat serum +0.1% Tween-20 in Tris-buffered saline (TBS)] and incubated overnight at 4 °C with primary antibodies specifically interacting with the antigen on the section. Slides were washed using phosphate buffer. The products of the interaction of primary antibodies with the antigen were revealed using the horseradish peroxidase conjugate specifically bound to secondary anti-species antibodies. The Novolink ™ Polymer Reagent Kit was used to detect bound primary antibodies (#RE7150, Leica, Newcastle upon Tyne, UK) followed by counter-staining with Mayer’s hematoxylin solution (LLC «BioVitrum», article: 05-002/S), dehydration in a graded ethanol series, and mounting with Vitrogel (LLC «BioVitrum», article 12-005). The result of the reaction was the development of the color of the brown chromogenic substrate. An aqueous solution of 3,3-diaminobenzidine tetrahydrochloride (DAB) was used to stain the immunohistochemical reaction product. Products of a positive immunohistochemical reaction were determined as brown staining of the membrane and/or cytoplasm of cells. For the negative control, the sections were subjected to standard immunohistochemical procedures without incubation with primary antibodies. The positive control was selected according to the specifications of the manufacturer.

For clinical specimens, we used mouse antibody (mAb) to cytokeratin 8 (cat# DB098-RTU, DB Biotech, Kosice, Slovakia, EU) for the detection of epithelial cells, mouse mAb to α-SMA (#ab7817, Abcam) for smooth muscle cells, rabbit mAb to CD34 (#ab81289, Abcam) for endothelial cells, rabbit mAb t to FABP4 (#ab219595, Abcam) for adipocytes, mouse mAb to CD68 (#ab783, Abcam), rabbit mAb to CD163 (#ab182422, Abcam), and rabbit polyclonal antibodies to CD206 (#ab64693, Abcam) for macrophages. Specimen microscopy was performed using the Leica microscope system, consisting of the Leica DM2500 microscope, Leica DFC290 video camera image microscopy (Leica, Wetzlar, Germany), and analysis software Image Scope M ver.1 (SMA, Moscow, Russia).

Statistical Analysis. All statistical analyses were performed with STATISTICA (Ver.12, StatSoft Inc., Tulsa, OK, USA) and Prism, version 6.0 (GraphPad Software Inc., San Diego, CA, USA). Due to the small size of the samples, the statistical analysis was carried out by nonparametric methods. The results were presented as a median with 25% and 75% percentiles—Me (Q1;Q3). Two groups were compared using the Mann–Whitney U-test, and three groups using the Kruskal–Wallis test; the differences were considered statistically significant at *p* < 0.05.

## 5. Conclusions

Therefore, according to the experimental finding, adipocytes should be absent in the uterine wall by the 12th sexual cycle after a full-thickness surgical incision. However, the presence of adipocyte clusters in cesarean scar indicated the disturbance of cell interactions in the uterine tissue. Moreover, significant differences in the numbers of CD206 and CD163 cells in adipocyte clusters between groups with and without PAS may be indirect evidence that uterine adipocytes affect the development of PAS.

## Figures and Tables

**Figure 1 ijms-24-15255-f001:**
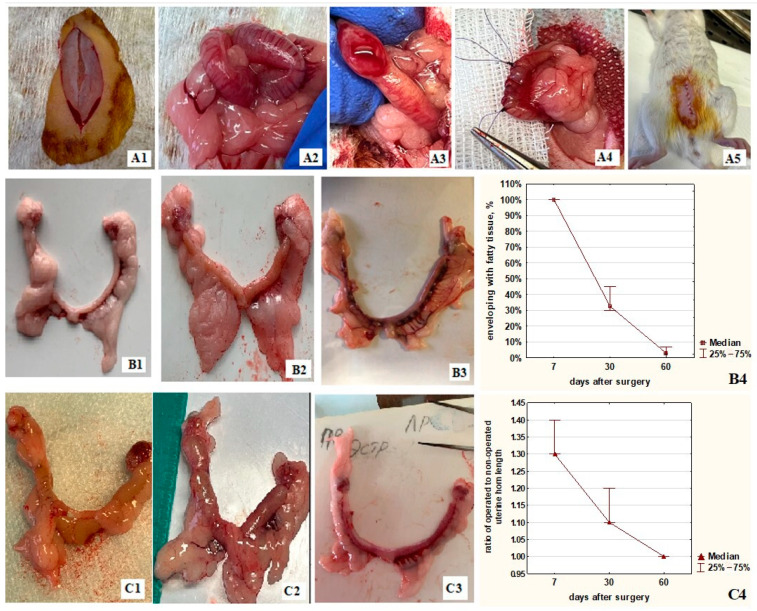
Gross picture of rat uterine horns after a full-thickness surgical incision. (**A1**–**A5**) Conducting a surgical incision of the uterine horn; (**B1**–**B3**) Fat tissue enveloping of the operated horn (left) on the 7th day (**B1**), 30th day (**B2**), and 60th day (**B3**); (**B4**) Covering the damaged area with fat tissue at different times after surgery; (**C1**–**C3**) The difference in length of the operated horn (left) compared to the intact one on the 7th day (**C1**), 30th day (**C2**), and 60th day (**C3**); (**C4**) The ratio of the operated horn length to the intact horn length at different times after surgery.

**Figure 2 ijms-24-15255-f002:**
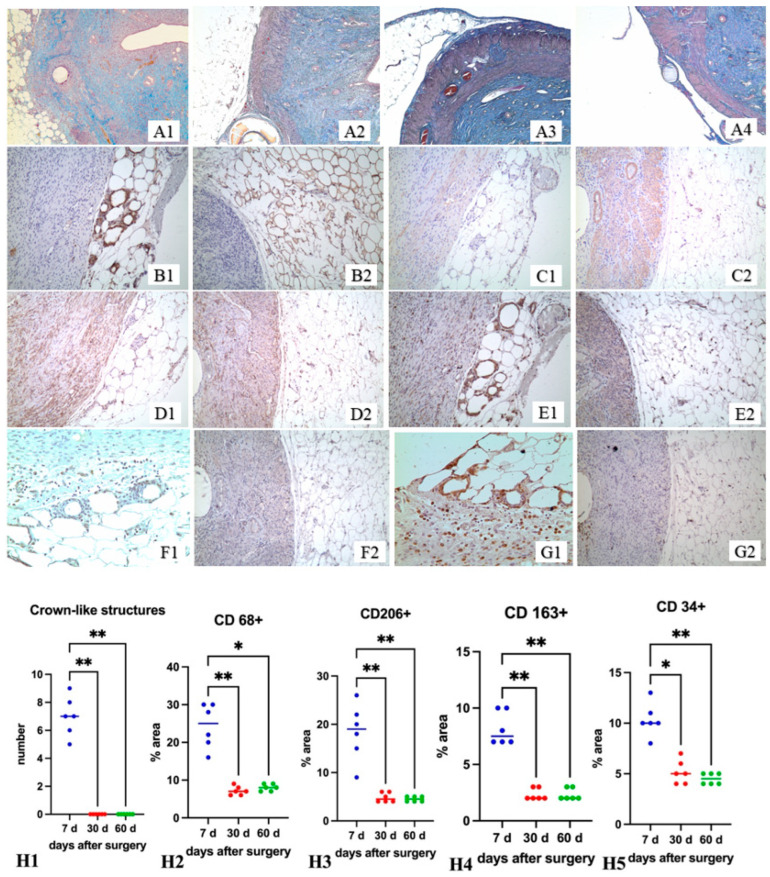
The healing area of the operated rat uterine horn. (**A**) ×50, Mallory staining; (**B**) ×100, immunohistochemical staining on FABP4; (**C**) ×100, immunohistochemical staining on α-SMA, Mayer’s hematoxylin staining; (**D**) ×100, CD34 immunohistochemical staining, Mayer’s hematoxylin staining; (**E**) ×100, CD68 immunohistochemical staining, Mayer’s hematoxylin staining; (**F**) ×200, CD163 immunohistochemical staining, Mayer’s hematoxylin staining; (**G**) ×200, CD206 immunohistochemical staining, Mayer’s hematoxylin staining; (**A1**,**B1**,**C1**,**D1**,**E1**,**F1**,**G1**) 7th day after surgery; (**A2**,**B2**,**C2**,**D2**,**E2**,**F2**,**G2**) 30th day; (**A3**,**A4**) 60th day; (**H1**) The crown-like structures in the area of attachment of adipose tissue to the uterine wall; (**H2**) CD68 cells in the adipose tissue attached to the uterine wall on the 7th, 30th, and 60th days after surgery; (**H3**) CD206 cells in the adipose tissue attached to the uterine wall on the 7th, 30th, and 60th day after surgery; (**H4**) CD163 cells in the adipose tissue attached to the uterine wall on the 7th, 30th, and 60th days after surgery; (**H5**) CD34 cells in the adipose tissue attached to the uterine wall on the 7th, 30th and 60th days after surgery; (*) *p* < 0.05; (**) *p* < 0.01; (●) The blue dot presents value for individual sample on the 7th day after the operation; (●) The red dot presents value for individual sample on the 30th day after the operation; (●) The green dot presents value for individual sample on the 60th day after the operation. The blue, red, green horizontal lines present the medians.

**Figure 3 ijms-24-15255-f003:**
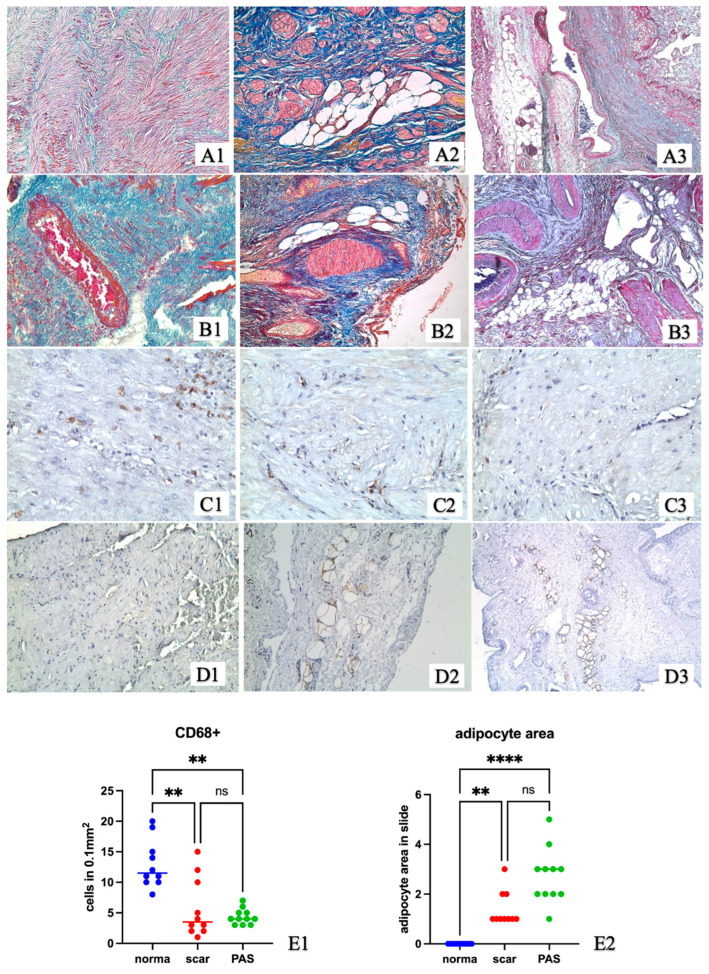
The uterine wall of women near the placenta. (**A**,**B**) Mallory staining; (**A1**,**B1**,**C1**,**D1**) Uterus without previous cesarean section and uterine scar; (**A2**,**B2**,**C2**,**D2**) Uterus with caesarian scar and non-complicated pregnancy; (**A3**,**B3**,**C3**,**D3**) Uterus with PAS; (**A1**,**A3**,**B3,D1**–**D3**) ×50; (**A2**,**B2, C2**,**C3**) ×100; (**C1**) ×200; (**C1**–**C3**) CD68 immunohistochemical staining, Mayer’s hematoxylin staining; (**D1**–**D3**) FABP4 immunohistochemical staining, Mayer’s hematoxylin staining; (**E1**) CD68 cells in comparison groups; (**E2**) Adipocyte clusters in comparison groups; norma—the control group of women without previous cesarean section and uterine scar; scar—the group of women with a cesarean scar and non-complicated pregnancy; PAS—a group of women with PAS; (**) *p* < 0.01; (****) *p* < 0.001; (ns) not significant; (●) The blue dot presents value for individual sample of the control group of women without previous cesarean section and uterine scar; (●) The red dot presents value for individual sample of the group of women with a cesarean scar and non-complicated pregnancy; (●) The green dot presents value for individual sample of a group of women with PAS. The blue, red, green horizontal lines present the medians.

**Figure 4 ijms-24-15255-f004:**
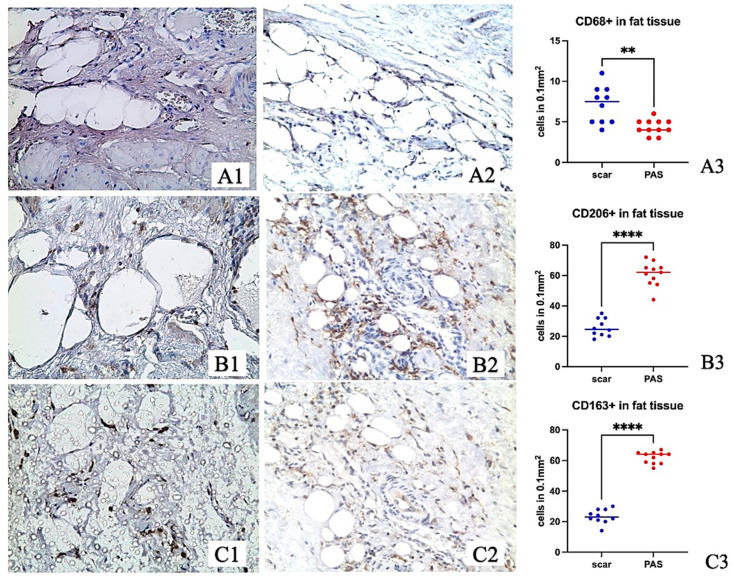
Adipocyte clusters in the uterine wall of pregnant women with scars in the case of women with non-complicated pregnancy and with PAS. ×200. (**A**) CD68 immunohistochemical staining, Mayer’s hematoxylin staining; (**B**) CD206 immunohistochemical staining, Mayer’s hematoxylin staining; (**C**) CD163 immunohistochemical staining, Mayer’s hematoxylin staining; (**A1**,**B1**,**C1**) Group with scar and non-complicated pregnancy; (**A2**,**B2**,**C2**) Group with PAS; (**) *p* < 0.01; (****) *p* < 0.001; (●) The blue dot presents value for individual sample of a group of women with a cesarean scar and non-complicated pregnancy; (●) The red dot presents value for individual sample of a group of women with PAS. The blue and red horizontal lines present the medians.

**Figure 5 ijms-24-15255-f005:**
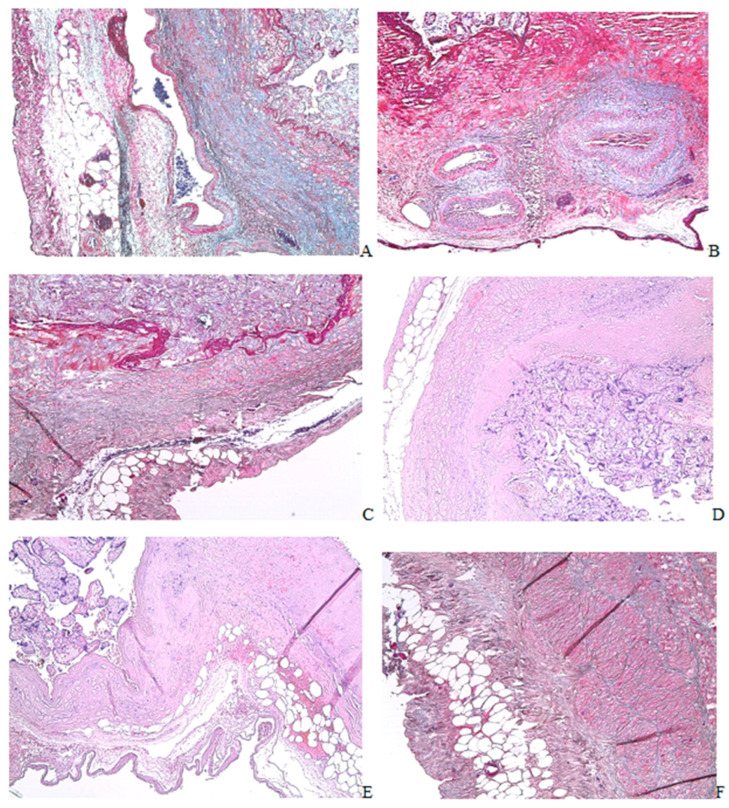
The uteroplacental region with abnormal invasion of placental villi (placenta increta), ×50; (**A**–**C**,**F**). Mallory stain; (**D**,**E**). Staining with hematoxylin and eosin.

**Figure 6 ijms-24-15255-f006:**
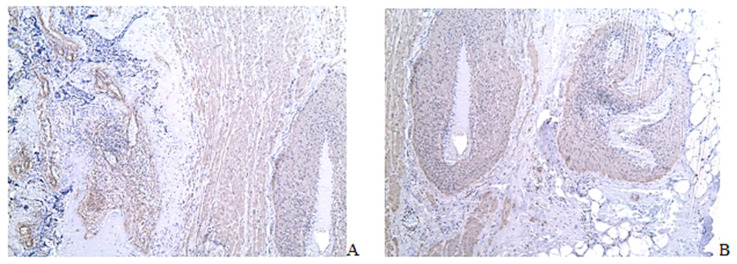
Vessels in the uterine wall with placental villi invasion. ×50. α-SMA immunohistochemical staining, Mayer’s hematoxylin staining; (**A**). The vessel in the scar zone with thickened muscle wall, disintegration, and dissociation of muscle fibers; (**B**). Continuation of the uterine wall of (**A**), vessels with thickened muscle wall, disintegration and dissociation of muscle fibers of the scar area, clusters of adipocytes in the perivascular space, and perimetrium zone.

**Figure 7 ijms-24-15255-f007:**
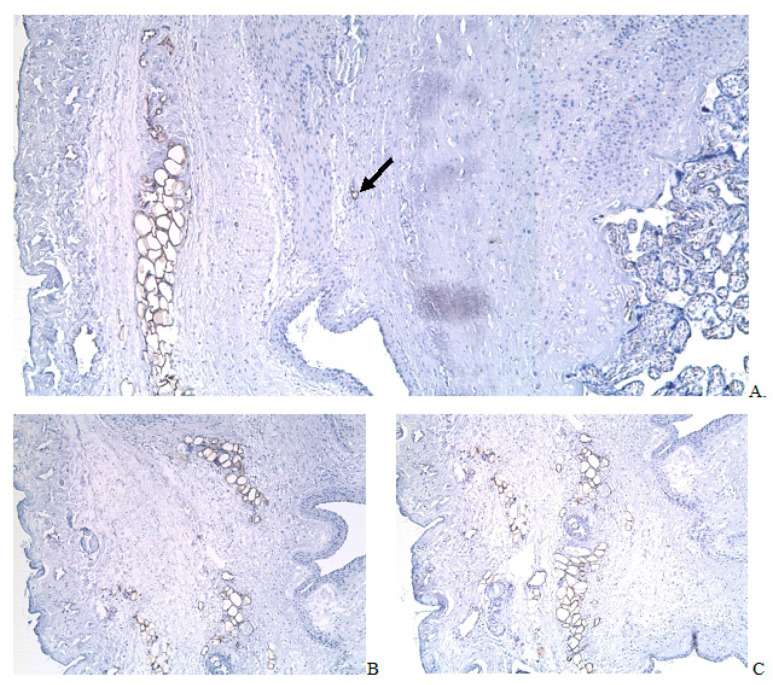
The uteroplacental region with PAS, ×50. FABP4 immunohistochemical staining, Mayer’s hematoxylin staining; (**A**) Villi invaded the myometrium in the scar area, а cluster of adipocytes; (**B**) Adipocyte clusters in the perivascular space; (**C**) Adipocyte clusters in the perivascular space (continuation of the uterine wall shown in (**B**)); black arrow indicates a single adipocyte.

**Figure 8 ijms-24-15255-f008:**
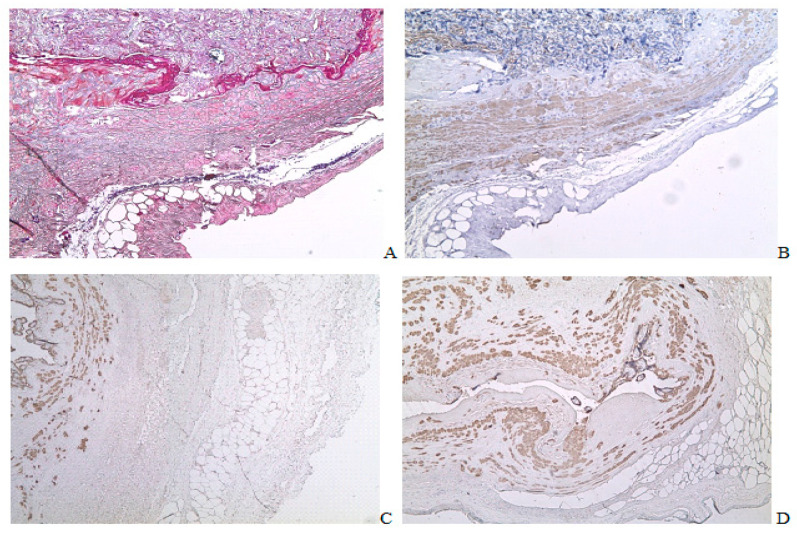
Adipocyte clusters arranged along the serosa in the uteroplacental region with placental villi invaded, ×50. (**A**)—Mallory staining; (**B**)—aSMA immunohistochemical staining, Mayer’s hematoxylin staining; (**C**,**D**)—Cytokeratin 8 immunohistochemical staining.

**Figure 9 ijms-24-15255-f009:**
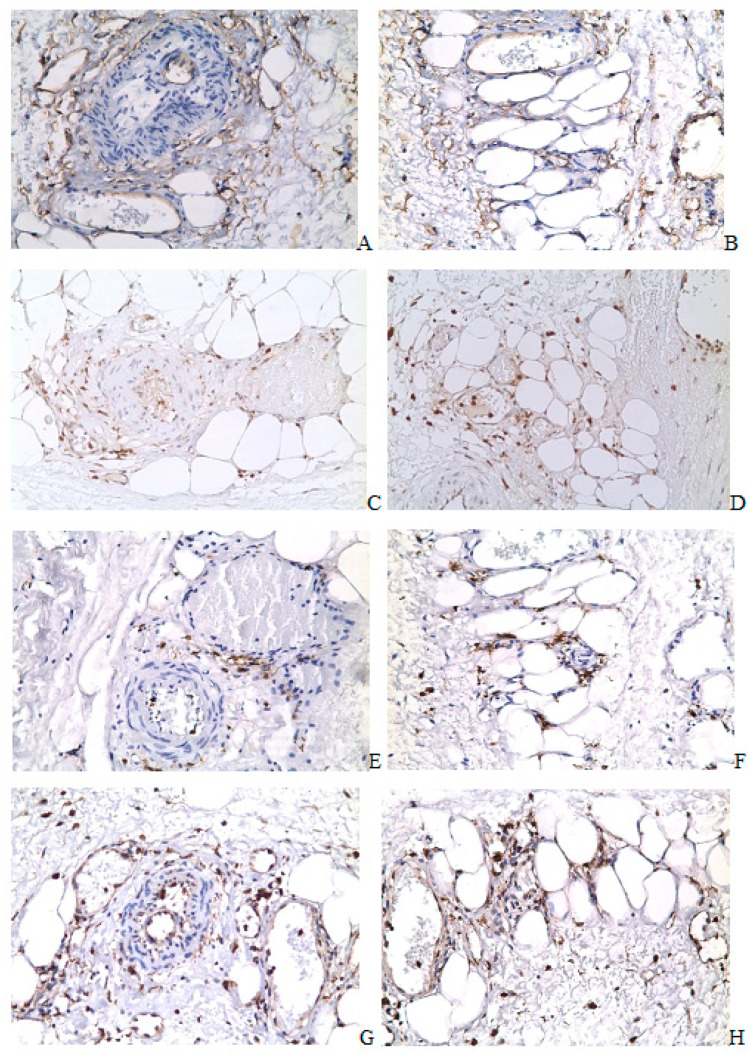
Adipocyte clusters located near the vessels in the uterine scar zone with pathological invasion of placental villi. Immunohistochemical staining with Mayer’s hematoxylin, ×200. (**A**,**C**,**E**,**G**)—Venous and arterial vessels inside the adipocyte cluster; (**B**,**D**,**F**,**H**)—Adipocyte cluster near the vessels; (**A**,**B**)—Staining for CD34; (**C**,**D**)—Staining for CD68; (**E**,**F**)—Staining for CD163; (**G**,**H**)—Staining for CD206.

**Figure 10 ijms-24-15255-f010:**
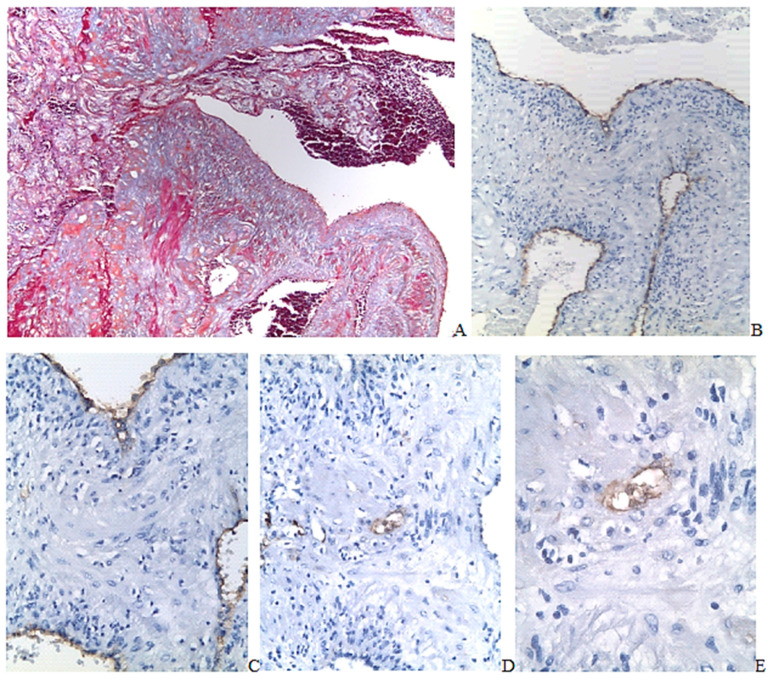
Vascular collectors of the uteroplacental region in the area of placental villus invasion. (**A**). Mallory staining, ×50; (**B**,**C**). CD34 immunohistochemical staining, ×100 (**B**), ×200 (**C**); (**D**,**E**). FABP4 immunohistochemical staining, ×200 (**D**), ×400 (**E**).

**Table 1 ijms-24-15255-t001:** Clinical characteristics of patients.

Group	Control (n = 10)		Scar (n = 10)		PAS (n = 11)	
	Me (Q1–Q3)	Min–Max	Me (Q1–Q3)	Min–Max	Me (Q1–Q3)	Min–Max
Age, years	29.5 (26;37)	25–41	35 (33;38)	27–41	35 (33;38)	29–42
Number of CS	1 (1;1)	1–1	2 (2;3)	2–4	2 (2;2)	2–3
Gestational age, weeks	39 (39;40)	39–40	39 (38.5;39.5)	38–40	34 (33;37)	27–38

## Data Availability

The data presented in this study are available on request from the corresponding author.

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
