# Peer review of "Adipocytes in the Uterine Wall during Experimental Healing and in Cesarean Scars during Pregnancy"

_ijms, 2023, doi:10.3390/ijms242015255_

Round 1

Reviewer 1 Report

To authors,

1.     Abstract is very long. It consists of approximately 800 words. The longer you write, the less clear the context becomes. Reduce the abstract volume at least by1/3. 

2.     Abstract; Please confirm whether the followings are doubled, “Statistical processing was carried out by nonparametric methods due to small sample size. The results were presented as a median with 25% and 75% percentiles - Me (Q1;Q3). Two groups were compared using the Mann-Whitney U-test,” 

3.     Please definitely describe your hypothesis. You only wrote the meaning that adipose cells might play some role in uterine scar healing but you did not state your hypothesis, especially the role of adipose cells in PAS. State you hypothesis. 

4.     Introduction: Incredibly long! What is the point? I guess the following is the point. “Adipose cells are reported to take part in tissue remodeling and thus healing process of the surgical scar. PAS is considered to occur from/in the previous cesarean scar and thus we hypothesized that adipose cells in the uterine cesarean scar-healing might some role in PAS-development. We attempted to determine whether this is the case and if yes, how. We performed this study to determine this mainly based on the histochemical study”. I am not confident whether this is the very point that you wish to state; however, you had better write things in this simple manner. If you wish to state something more, please state it in discussion. Once again, state your hypothesis more definitely, clearly, and in a reader-friendly manner  (aim, question). 

5.     Results and Methods: OK but too long. 

6.     Discussion. Very long. Please summarize the role of adipose cells playing in the development of PAS. You wrote it here and there. Please summarize!

7.     General: The data might be of some use. Please shorten the manuscript. English is awkward and wrong here and there. Please recheck English. 

Polish is necessary. 

Author Response

Dear Reviewer, thank you very much for taking the time to review this manuscript. Your comments were very helpful in improving our manuscript.

Reviewer 2 Report

Esteemed authors and Editorial team,

In my opinion this article requires major revision. The study consists of two parts which should not be associated: an experimental study performed on rats and a retrospective human study performed on patients with prior caesarean delivery. I suggest they should be managed as two separate manuscripts.

Secondly, the order of the sections in the text must be reviewed as well as the quality of presentation: the introduction is too long, some aspects require being moved to discussions, the materials and methods section is incomplete (the authors do not specify how scar tissue was sampled). Some of the conclusions are not reflected by the results because of the limited number of patients included in the analysis.

The references need to be updated and extended.

Major revision of the English language is required, in my opinion.

Author Response

(The authors gave the same response as above.)

Reviewer 3 Report

The article investigates the healing process of the uterine wall after a thorough surgical incision in rats. The study examines the role of adipose tissue in the healing process and the dynamics of the healing process over time. The study finds that adipose tissue plays a significant role in the healing process, with adipocytes attaching to the healing zone and forming crown-like structures. The study also finds that the healing process is dynamic, with the length of the operated horn recovering by the 30th day and completely recovering by the 60th day. The article provides gross pictures and histological examinations of the uterine horns at different time points after surgery to support the findings.

The article can be further enhanced by adding/answering comments below.

1.     The study included a small sample size of only 31 participants, which may limit the generalizability of the findings.

2.     Limited demographic information: The article provides limited demographic information about the participants, such as age and gestational age. Other factors, such as ethnicity, socioeconomic status, and medical history, are not reported, which may affect the interpretation of the results.

3.     The inclusion and exclusion criteria for the study groups may have introduced bias. For example, the criteria for inclusion in the group with the first CS were operative delivery, such as the incorrect position of the fetus, anatomical narrow pelvis, fetal macrosomia, the presence of a uterine scar, and eye pathology in women. The exclusion criteria for all groups included infectious diseases, autoimmune pathologies, metabolic disturbances, obesity of the second and third degree, and diabetes mellitus type 2.

4.     The study does not mention controlling for potential confounding factors that could influence the outcomes, such as the use of medications, lifestyle factors, or other medical conditions.

5.     The article does not provide a detailed description of the methods used for data collection and analysis. This makes it difficult to assess the validity and reliability of the study's findings.

6.     The article does not compare its findings with those of other studies in the field. This makes it difficult to determine whether the results are consistent with previous research or if they represent new insights.

7.     The article does not discuss potential biases that may have influenced the study's findings, such as observer bias, measurement bias, or recall bias. This limits the ability to assess the validity of the results.

8.     The article does not mention whether the researchers were blinded to the participant's group allocation during data collection and analysis. Blinding is an important aspect of study design to minimize potential biases and ensure the validity of the results.

9.     The article provides limited information on the statistical methods used to analyze the data, making it difficult to assess the robustness of the findings.

10.  The article does not discuss the potential clinical implications of the study's findings, which limits the applicability of the results to real-world settings.

good

Author Response

(The authors gave the same response as above.)

Round 2

Reviewer 2 Report

Accept in present form.